# Genome-Wide Identification and Expression Analysis of HSP70 Gene Family Under High-Temperature Stress in Lettuce (*Lactuca sativa* L.)

**DOI:** 10.3390/ijms26010102

**Published:** 2024-12-26

**Authors:** Qian Wang, Wenjing Sun, Yipei Duan, Yikun Xu, Huiyu Wang, Jinghong Hao, Yingyan Han, Chaojie Liu

**Affiliations:** 1College of Plant Science and Technology, Beijing University of Agriculture, Beijing 102206, China; wangq000213@163.com (Q.W.); wjs0618@126.com (W.S.); d18336723558@163.com (Y.D.); wanghy@bua.edu.cn (H.W.); haojinghong2013@126.com (J.H.); hyybac@126.com (Y.H.); 2College of Food Science and Engineering, Shandong Agricultural University, Taian 271018, China; whitedragon13@163.com

**Keywords:** gene expression, genome-wide identification, high-temperature stress, HSP70 gene family, lettuce

## Abstract

The heat shock protein 70 (HSP70) family plays an important role in the growth and development of lettuce and in the defense response to high-temperature stress; however, its bioinformatics analysis in lettuce has been extremely limited. Genome-wide bioinformatics analysis methods such as chromosome location, phylogenetic relationships, gene structure, collinearity analysis, and promoter analysis were performed in the *LsHSP70* gene family, and the expression patterns in response to high-temperature stress were analyzed. The mechanism of *LsHSP70-19* in heat resistance in lettuce was studied by virus-induced gene silencing (VIGS) and transient overexpression techniques. The results showed that a total of 37 *LsHSP70* genes were identified by the Hidden Markov Model (HMM) and Protein Family Database (Pfam). These 37 *LsHSP70* genes were classified into groups A, B, C, and D by phylogenetic relationships. They were mainly localized on seven chromosomes except for chromosome 3; gene structure analysis showed that *LsHSP70* contained 1–9 exons, and the protein structure domains of genes in the same group were highly conserved. The covariance analysis showed that nine pairs of *LsHSP70* genes existed between *LsHSP70* members, and lettuce *LsHSP70* and sunflower *HaHSP70* had been more conserved in the evolutionary process. The promoter analysis showed that there were a large number of cis-acting elements related to phytohormones, growth, development, stress, and light response in *LsHSP70*. In addition, the results of the expression pattern analysis for all *LsHSP70* genes under high-temperature stress showed that 28 out of 37 *LsHSP70* genes were able to respond to heat stress, and only *LsHSP70-8*, *LsHSP70-14*, *LsHSP70-19*, *LsHSP70-23*, and *LsHSP70-24* were able to respond rapidly to heat stress (2 h). The expression of *LsHSP70-19* was higher at different periods under high-temperature stress; the overexpression of *LsHSP70-19*, the plant fresh weight, and the root weight were better than the control (CK); and the heat resistance was better. These results suggest that *LsHSP70-19* may play an important role under high-temperature stress in lettuce.

## 1. Introduction

High temperature is one of the unfavorable environmental factors in crop growth and development, and high-temperature stress can cause irreversible damage to plants when the ambient temperature is persistently higher than the optimum temperature for plant growth. Heat shock proteins (HSPs), produced as a result of high-temperature stress, are present in almost all organisms and are highly conserved, with low levels under normal conditions, but they are synthesized in large quantities in a short period of time when stimulated by unfavorable conditions in the external environment [1]. HSPs can be categorized as HSP100, HSP90, HSP70, HSP60, and small-molecule heat shock proteins (sHSPs) based on their molecular weight [2]. HSPs mainly act as molecular chaperones during plant growth and development to assist in various types of protein folding, ensure the correct folding of proteins, avoid damage to proteins, maintain the stability of the plant internal environment, and play an important role in plant resistance to adversity stress [3].

HSP70 is usually categorized as HSP70, HSC70 (heat shock cognate 70), mtHSP70, or BiP/Grp78 (binding immunoglobulin protein) [4]. The HSP70 gene family generally has two typically conserved structural domains. Genes are enriched on chromosome 9. The C-terminal end consists of two subregions, the SBD and the C-terminal “lid”, which assists in the substrate binding of the SBD, and it binds, recognizes, and processes substrate proteins, while the NBD at the N-terminal end is a highly conserved region, which binds and hydrolyzes ATP [5,6].

HSP70 plays an important role in plant growth, development, and resistance to various stresses [7]. Under various biotic and abiotic stresses, the expression level is often higher, and the anti-stress ability of plants is improved. The chili *CaHSP70-2* gene was overexpressed in *Arabidopsis thaliana* to enhance its tolerance to high temperature [8]. The overexpression of Arabidopsis cytoplasmic *HSC70-1* effectively enhanced heat tolerance in transgenic plants [9]. The overexpression of tobacco *NtHSP70-1* elevated plant tolerance to drought stress [10]. The expression of *TaHSC70* in the leaves of yellow-flowering seedlings was increased after heat stress in wheat seedlings at 40 °C [11]. It was shown that when the thermosensitive mutant carrying the T-DNA insertion HSP70 gene *OsHSP70CP1* grew at the high temperature of 40 °C, the seedling leaves showed severe yellowing, and the plant phenotype was normal at 27 °C [12]. In addition, HSP70 is involved in the high-temperature stress response in plants such as grape and chrysanthemum [13,14]. Currently, the involvement of HSP70 under high-temperature stress has been reported in Arabidopsis (*Arabidopsis thaliana* L.) [15], spinach (*Spinacia oleracea* L.) [16], chili pepper (*Capsicum annuum* L.) [8], wheat (*Triticum aestivum* L.) [11], rice (*Oryza sativa* L.) [17], potato (*Solanum tuberosum* L.) [7], soybean (*Glycine max* L.) [18], and cotton (*Gossypium hirsutum* L.) [19] regarding the identification and expression analysis of the HSP70 gene family.

Lettuce is a leafy vegetable, widely grown around the world, that prefers coolness and does not tolerate high temperatures [20]. It is rich in nutrients and minerals. Like some Asteraceae crops, it is susceptible to high temperatures during growth and development with lower quality and yield [14]. The identification of the HSP70 gene family in lettuce and its functional validation have not yet been reported. The physicochemical properties, phylogenetic relationships, gene structure, and role elements in the promoter region of *LsHSP70* were analyzed. The expression pattern of *LsHSP70* under heat stress for 2 h and 4 days were studied, the *LsHSP70-19* genes that responded rapidly to heat stress were screened out, and we initially verified the positive effect on lettuce heat tolerance by VIGS and transient overexpression techniques. The positive effect on heat tolerance in lettuce was initially verified. A preliminary framework for exploring lettuce HSP70 genes was constructed. We provide a theoretical basis for the subsequent analysis of the role of *LsHSP70* lettuce under high-temperature stress.

## 2. Results

### 2.1. Identification, Chromosomal Localization and Physicochemical Properties of HSP70 Family Members in Lettuce

Based on the structural sequence of Arabidopsis HSP70, 37 HSP70 genes in lettuce were mined. The conserved structural domains of all the putative 37 genes by Pfam and CDD databases were identified. Eventually, 37 HSP70 genes were identified in lettuce (Table 1), which were named *LsHSP70-1* to *LsHSP70-37*. The coding sequences (CDS) ranged from 1948 bp to 6494 bp, protein lengths were between 598 and 701 amino acids, and isoelectric points ranged from 5.07 to 8.48, among which 13 genes with pI > 7, and the rest were acidic (pI < 7). These 37 genes were from seven chromosomes of lettuce, of which 10 *LsHSP70* genes were enriched on chromosome 9 (Figure 1). Among these 37 genes *LsHSP70-33* had the lowest molecular weight (MW), and *LsHSP70-15* had the highest MW. Subcellular localization predictions showed that *LsHSP70* genes could be localized in the nucleus, mitochondria, chloroplasts, and endoplasmic reticulum, and 14 of them were predicted to be localized in chloroplasts, which was the organelle that appeared with the highest frequency among the predicted locations.

### 2.2. Phylogenetic Relationship Analysis of LsHSP70

The phylogenetic tree was constructed by HSP70 from *Arabidopsis thaliana*, rice, and lettuce (Figure 2). *LsHSP70* are categorized into four groups: A, B, C, and D. Group A had the highest number of 21 *LsHSP70* genes, while group C contained only *LsHSP70-17* and *LsHSP70-18*, and groups B and D had nine and five *LsHSP70* genes each. In addition, there were no *AtHSP70* genes in group C, and the *OsHSP70* genes were also distant from the *LsHSP70* genes, which may indicate that this class of *LsHSP70* genes was distantly related to the HSP70 genes of Arabidopsis and rice.

### 2.3. Gene Structure and Motif Localization Analysis of LsHSP70

Gene structure prediction showed (Figure 3B) that the number of exons in *LsHSP70* genes ranged 1–9, and all 19 genes in group A contained two exons except for *LsHSP70-1* and *LsHSP70-12*, which had one and nine exons, respectively. All genes in group B contained two exons except for *LsHSP70-14*. All genes in group C contained only two exons. The genes in group D contained a high number of exons, containing six or eight exons.

The conserved motifs of *LsHSP70* were analyzed by MEME 5.5.7 software, and a total of 15 different motifs were detected (Figure 3C). The vast majority of the genes in group A contained 14 or 15 motifs. All nine *LsHSP70s* in group B contained 15 motifs. *LsHSP70-17* and *LsHSP70-18* in group C contained 13 and 14 motifs, respectively. Group D contained the fewest motifs, and *LsHSP70-13* contained only seven motifs. All 37 *LsHSP70s* contained Motif1, Motif2, Motif3, Motif6, Motif8, and Motif11. This indicates that these six motifs may play a central role in the *LsHSP70* protein domain. They are not only involved in the basic molecular chaperone function of HSP70, and they may be involved in signal transduction, protein stability and cell stress response.

### 2.4. Covariance Analysis of LsHSP70

The covariance analysis reveals the evolutionary mechanism of the gene family of the species. Among the 4479 covariate pairs in lettuce (Figure 4), nine pairs of *LsHSP70* genes were identified to be in a covariate relationship. This suggests that these nine pairs of genes were formed through large fragment duplication. A higher number of genes with covariance existed on chromosomes 8 and 9 with the *LsHSP70-20* covariant with both the *LsHSP70-21*, *LsHSP70-28*, and *LsHSP70-36*, and *LsHSP70-36* covariant with *LsHSP70-8*, *LsHSP70-20*, and *LsHSP70-28*.

In addition, the homology of the *LsHSP70* gene family was further investigated, the covariances at the gene level in lettuce HSP70, Arabidopsis HSP70, and sunflower HSP70 were analyzed (Figure 5). There were three covariate pairs belonging to *LsHSP70* in lettuce and Arabidopsis, which were from chromosome 5 in Arabidopsis and chromosomes 2 and 9 in lettuce, respectively. In comparison with sunflower, there were nine covariate pairs, but again, most of the covariate pairs were on chromosome 9 of lettuce. This suggests that the *LsHSP70* gene may be more closely related to sunflower.

### 2.5. Cis-Acting Element Analysis of LsHSP70

The cis-acting elements of the upstream 2000 bp promoter region of 37 *LsHSP70* genes were analyzed (Figure 6). A total of 34 regulatory elements were identified as belonging to four categories. These categories included phytohormones, growth and development, stress response, and light response. The highest number of action elements was related to light response, and the smallest number of elements was related to the growth and development category. Among the phytohormone response elements ABRE, CGTCA and TGACG appeared most frequently as abscisic acid (ABA) and methyl jasmonate (MeJA) response elements, respectively. Among the growth and development-related elements, O2-site and CAT-box appeared with the highest frequency, which were maize protein metabolism and meristematic tissue expression response elements, respectively. The ARE and MBS stress response elements responding to anaerobiosis and drought appeared with higher frequencies. The light response-related elements appeared with similar frequencies, and they were also more evenly distributed among individual *LsHSP70* genes. The results suggest that a series of elements with higher frequencies may be closely related to the role of *LsHSP70* in different environments.

### 2.6. Analysis of the Expression Pattern of LsHSP70 Under High-Temperature Stress

The expression levels of the *LsHSP70* gene family in response to high-temperature stress were analyzed by transcriptome data (Figure 7). The expression levels of the vast majority of *LsHSP70* genes were low at the early stage of stress (2 h of stress). *LsHSP70-19*, *LsHSP70-23*, *LsHSP70-24*, *LsHSP70-8*, and *LsHSP70-14* exhibited faster responses to heat stress. According to the phylogenetic tree, four of the five *LsHSP70s* belonged to group B, while *LsHSP70-19* belonged to group A, and *LsHSP70-19* was differentially expressed at its gene level. The expression levels of the majority of the genes were enhanced at the 4th day with the prolongation of stress. The expression levels of seven genes were decreased after the 4th day under high-temperature stress, *LsHSP70-36*, *LsHSP70-9*, *LsHSP70-13*, *LsHSP70-10*, *LsHSP70-4*, *LsHSP70-6*, and *LsHSP70-26*. However, the expression of *LsHSP70-36* was the weakest, and its expression was smaller throughout the stress period. Most of these seven genes belonged to group A. This suggests that *LsHSP70* in group A may negatively regulate the tolerance of lettuce under high-temperature stress. Therefore, *LsHSP70-19*, a more prominent expression pattern in group A, was selected for further study.

### 2.7. Role of LsHSP70-19 Under High-Temperature Stress

The role of *LsHSP70-19* gene on lettuce heat tolerance was further explored. The gene for silencing as well as overexpression was analyzed (Figure 8). We cloned 543 bp and full-length fragments from the CDS region of *LsHSP70-19* into pTRV2 and pRI101 vectors, respectively. After Agrobacterium was mediated and validated, the constructs pTRV2-*LsHSP70-19* and pRI101-*LsHSP70-19* were successfully transferred into lettuce and expressed.

The expression in the silencing lettuce was lower than that in the control (CK), and the expression in the overexpression plants was higher than that in CK. The leaves of silenced lettuce were elongated, and the total fresh weight and shoot fresh weight were significantly lower than CK, while the leaves of overexpression showed no significant change in phenotype with the CK, but the total and shoot fresh weight were significantly higher than CK (Table 2). The above results indicate that the overexpression of *LsHSP70-19* can improve the heat resistance of lettuce.

## 3. Discussion

Lettuce is a typical heat-sensitive vegetable, it is susceptible to high-temperature stress in summer during annual production, and its quality and yield are decreased. Few studies have been found on the HSP70 family in lettuce; therefore, the HSP70 family was characterized using bioinformatics tools and its expression pattern and role were explored under high-temperature stress.

The 37 HSP70 genes were found in lettuce, a significant difference from the number of 18 HSP70 genes in Arabidopsis [15], which may be due to species differences. The 61, 21, and 33 HSP70 genes were found in soybean [18], pumpkin [21], and grape [13], respectively, and their HSP70 genes were not exactly equal in number, either. A total of 113 HSP70 genes were identified in *Gossypium hirsutum*, *G. barbadense*, *G. arboreum*, and *G. raimondii*. Among them, there were 33 HSP70 genes in *G. arboreum* and 32 HSP70 genes in *G. barbadense* [22]. Even for the same family, the number of homologous gene families varies among species due to the size of the genome number and the direction of evolutionary selection [18,23].

The *LsHSP70* family in terms of physicochemical properties, phylogenetic relationships, gene structure, and cis-acting element analysis of promoters was analyzed. The *LsHSP70* gene family was distributed individually or in clusters on seven chromosomes of lettuce, and *LsHSP70* was absent from chromosome 3 and chromosome 7; similarly, none of the chromosomes 3, 9, 13, and 16 of the rowan tree distribute HSP70 family members [24]. No HSP70 gene family members were detected on chromosomes 1 and 12 of *Gossypium raimondii* [25]. Among the seven chromosomes of cucumber genome, only five chromosomes had 12 *CsHSP70* genes, and no *CsHSP70* genes were distributed on chromosomes 1 and 6 [26]. The evolutionary process of the plant genome structure is relatively conservative, which may lead to the retention of the HSP70 gene on specific chromosomes, while other chromosomes are not distributed.

Phylogenetically closely related HSP70 genes have the same or similar subcellular localization and gene structure [22]. Most of the genes in group A were localized in mitochondria; all of the genes in group C were localized in the endoplasmic reticulum and nucleus. However, the genes in group D were localized only in mitochondria or chloroplasts. The subcellular localization prediction of *Gossypium raimondii* showed that it was located in the cytoplasm, endoplasmic reticulum and mitochondria [25]. Comparing the homologous genes of the lettuce HSP70 gene family in rice and *Arabidopsis*, it can be found that they have similar subcellular localization [15,17].

According to motif analysis, the 21 *LsHSP70* genes contained 1–15 motifs, and the vast majority of genes in group A contained 14 or 15 motifs. All *LsHSP70* in group B contained 15 motifs. *LsHSP70-17* and *LsHSP70-18* in group C contained 13 and 14 motifs, respectively. Group D contained 13 motifs. In potato, the number and location of motifs of *StHSP70* genes in the same group were more similar [7]. The C-terminal motifs are more conserved than the N-terminal motifs. Unique characteristic motifs exist among different groups. This indicates their evolutionary relationships and different functional divisions [1].

In addition, according to the gene structure analysis, the number of introns in group A does not exceed two except for *LsHSP70-12*. In addition to *LsHSP70-1* and *LsHSP70-12* containing one and nine exons, the remaining *LsHSP70* genes contain two exons. Group B genes had one or no introns, except that *LsHSP70-14* contained only one exon, while the rest contained two exons. Group C contained only two exons. Group D contained six or eight exons. This is consistent with the findings in other species. *AtHSP70* genes with a similar number of exons in *Arabidopsis thaliana* are located in the same organelle [15], and the *GmHSP70* gene close to soybean has a similar number of exons [18]. The gene structure will determine the function of the gene to a certain extent. The presence of introns and rapid gene expression are controversial. Because there are few or no introns in plants, it can enhance the immediate activation of gene expression [27]. The genes of groups A and B can respond faster to heat stress compared to groups C and D.

Gene duplication is a common mechanism of gene family expansion, and it increases genetic novelty during plant evolution. It has important implications for genome evolution in plants [28]. Covariance analysis can further demonstrate the replicative relationships of gene families within species and evolutionary similarities between species. Within lettuce, nine gene pairs were collinear, three pairs of genes were collinear with *Arabidopsis thaliana*, and nine pairs of genes were collinear with sunflower. The sunflower belongs to the same family of Asteraceae crops as lettuce, and the two species have more direct genes between them and are more closely related than lettuce and Arabidopsis. However, *LsHSP70-28* was covariant with both Arabidopsis and sunflower, suggesting that this gene’s function is conserved across species. Among the 10 *StHSP70* genes in potato, three pairs of tandem genes and two pairs of segmented repeat genes were found [7]. The collinearity analysis of the pumpkin HSP70 gene family showed that there were nine pairs of gene fragment repeats [21]. In the cucumber genome, two *CsHSP70* genes formed a fragment repeat [26]. Therefore, species-specific duplication or deletion during evolution may lead to differences in the number of HSP70 family genes between different species.

Cis-acting elements play an important role in gene regulation and expression within promoter regions, and well-regulated promoter cis-acting elements are urgently needed for transgenic plant engineering to achieve controlled expression levels of target genes [29]. An analysis of promoter cis-acting elements of the 37 *LsHSP70* genes revealed that phytohormone-responsive and light-responsive elements were the most abundant. A majority of these *LsHSP70*-responsive elements were associated with ABA, MeJA, and an anaerobic environment. ABRE plays an important role in signaling, ABA activation, and resistance in Arabidopsis [30]. The cis-acting elements indicate that the *LsHSP70* gene family plays a key role in responding to different environmental changes such as phytohormones, drought, heat, and light.

The role of *LsHSP70* under high-temperature stress was investigated; the expression of this gene family was analyzed under heat stress (2 h of stress) and continuous heat stress (4 days). During heat stress, a small fraction of *LsHSP70* was able to rapidly respond to high-temperature changes and enhance their expression with a more than 40-fold increase in the expression of *LsHSP70-19*. The role of *LsHSP70-19* in heat tolerance in lettuce under high-temperature stress was initially verified by VIGS and overexpression After 1 week under high temperature, the fresh weight and root weight of the silenced plants were lower than CK, and the expression of *LsHSP70-19* was also lower than CK. After overexpressing *LsHSP70-19*, the heat tolerance of lettuce was improved, and *LsHSP70-19* played a positive regulatory role under high-temperature stress. In Arabidopsis, the expression of *AtHSP70* genes in Arabidopsis seedlings was significantly enhanced after 2 h of heat stress [15], and the expression levels of four *OsHSP70* genes in rice were also markedly enhanced after 3 h of high heat stress (42 °C) [31]. This suggests the existence of a number of HSP70 genes that can rapidly respond to high-temperature stress and enhance their expression levels. However, in persistent heat stress, the expression levels of most of the genes increased with the prolongation of stress time, and only a few of them remained unchanged or decreased after heat treatment. This suggests that there may be a functional division of labor among *LsHSP70* genes in response to heat stress.

## 4. Materials and Methods

### 4.1. Plant Materials and Treatments

The plant material used was heat-sensitive lettuce cultivar ‘Beisansheng No. 3’ by the Beijing University of Agriculture. The cultivate conditions were incubated at 20 ± 2 °C (day)/13 ± 2 °C (night), with a photoperiod of 14 h (day)/10 h (night), relative humidity of 60 ± 5%, and light intensity of 12,000 Lux. Lettuce was treated in 5 leaves. The treatment conditions were room temperature (CK) 20 ± 2 °C (day)/13 ± 2 °C (night), high temperature (H) 33 ± 2 °C (day)/25 ± 2 °C (night), and the rest of the conditions remained the same. Samples of lettuce leaves were taken at 2 h and 4 days of high-temperature stress with each treatment replicated at least 6 times.

### 4.2. Identification and Physicochemical Properties of HSP70 Family Members in Lettuce

The whole genome sequence of lettuce was downloaded from the Ensemble Plants database (https://plants.ensembl.org/index.html; accessed on 11 July 2022) as a basis to establish a local protein database. Overall, 18 HSP70 genes of *Arabidopsis thaliana* were obtained, and the constructed lettuce database was searched with BLASTP by the structural sequence of *Arabidopsis thaliana* as a query benchmark. The genes with an E-value of <0.05 were selected after deleting duplicates. The preliminarily screened lettuce HSP70 genes were subsequently confirmed by HMM and the Pfam database to confirm the presence of the HSP70 structural domain in each of them. Eventually, a total of 37 HSP70 genes were confirmed as *LsHSP70* genes in lettuce.

The ProtParam tool (https://web.expasy.org/protparam; accessed on 30 November 2022) was used to predict the physical and chemical properties of lettuce HSP70, including the number of amino acids (aa), the isoelectric point (pI), and the molecular weight (MW) of *LsHSP70*, and the results are demonstrated in Table 1.

Cell-PLoc 2.0 was used to predict subcellular localization (http://www.csbio.sjtu.edu.cn/bioinf/Cell-PLoc-2; accessed on 5 December 2022). HSP70 members for the remaining species were obtained from the corresponding plant databases, and data for each species were downloaded from Phytozome V13 (https://phytozome-next.jgi.doe.gov; accessed on 11 July 2022).

### 4.3. Chromosomal Localization of LsHSP70 and Construction of Phylogenetic Tree

By querying the annotation file of lettuce genome, the chromosome location and direction of HSP70 were obtained, and the gene density in chromosome was obtained by TBtools windows-x64_1_113. The chromosome location was visualized, and the chromosome location of 37 *LsHSP70* genes was completed. The construction of the phylogenetic tree was performed by MEGA 11.0 software. ClustalW 2.1 was utilized for multiple sequence comparison of lettuce, *Arabidopsis thaliana*, and rice. The phylogenetic tree was constructed by the neighbor-joining (NJ) method based on 1000 bootstrap replicates, and the output phylogenetic tree was finally colored and beautified using the iTOL website (https://itol.embl.de; accessed on 15 October 2021).

### 4.4. Gene Structure and Motif Analysis of LsHSP70

The 37 CDS of *LsHSP70* were mapped to the gene structure of lettuce HSP70 by TBtools windows-x64_1_113 software. The conserved motifs of *LsHSP70* were identified using a MEME tool (https://meme-suite.org/meme/info/status?service=MEME&id=appMEME_5.5.11676975855714812268290; accessed on 18 October 2021). A total of 15 motifs were selected, the characteristics of each motif were retained, and finally the motifs of 37 *LsHSP70* genes were mapped by TBtools windows-x64_1_113.

### 4.5. Analysis of Covariance

The established database and TBtools windows-x64_1_113 software were used to obtain the required files such as the chromosome skeleton, gene density, and the location of *LsHSP70* genes in lettuce. Circos plots were used to visualize the homology between *LsHSP70* genes. The collinearity analysis of genes among lettuce, *Arabidopsis thaliana* and sunflower was performed to analyze the homologous relationship between their HSP70s. The whole genome and annotation information of sunflower was from the phytozome database (https://phytozome-next.jgi.doe.gov; accessed on 7 February 2023).

### 4.6. Analysis of Promoter Action Elements of LsHSP70 Gene

A 2000 bp region upstream of *LsHSP70* was selected in the established lettuce database, and the cis-acting elements were identified and analyzed in this region. Data on cis-acting elements were from the PlantCARE online database (https://bioinformatics.psb.ugent.be/webtools/plantcare/html/; accessed on 20 October 2021). The identified elements were visualized by TBtools windows-x64_1_113.

### 4.7. Expression Analysis of LsHSP70-19 Under High-Temperature Stress

Fresh lettuce leaves were selected as materials for RNA extraction. The sample was filled in a 2 mL reagent tube containing 3 steel balls with a diameter of 2 mm. The tubes were labeled with the sample names and stored at −80 °C. Total RNA was extracted according to the Quick RNA Isolation Kit (Quick RNA Isolation Kit, Huayueyang, Beijing, China). The first cDNA strand was synthesized by a reverse transcription kit (Tiangen, Beijing, China). The resulting cDNA was used as a template for quantitative real-time polymerase reaction (qRT-PCR) with 18S as the internal reference gene. qRT-PCR was performed by TB Green Premix Ex TaqTM II (TaKaRa, Beijing, China) on a CFX real-time PCR detection system instrument (Bio-Rad Laboratories, Hercules, CA, USA). The internal reference gene was 18S, and the reaction system was 20 μL, including 9 μL TB Green Premix Ex TaqTM II (TaKaRa), 7 μL ddH2O, 1 μL forward primer, 1 μL reverse primer and 2 μL cDNA template. Expression was calculated by the 2^−ΔΔCt^ method, and three technical replicates were performed for each sample. The expression levels of *LsHSP70* were extracted from previous transcriptome data.

### 4.8. Construction of VIGS Gene Silencing and Overexpression Vector

The 543bp sequence of the CDS region of *LsHSP70-19* was amplified by a VIGS-*LsHSP70-19*-F/R primer and pRI101-*LsHSP70-19*-F/R primer (Appendix A). The pTRV2 vector was linearized by KpnI, and the pRI101 vector was linearized by BamHI and NdeI. A homologous recombination of amplified bands and linearized vectors was performed using a ClonExpressII One Step Cloning Kit (Vazyme Biotech, Beijing, China). The recombinant plasmids were transferred into DH5α competent cells, sequenced correctly, and then transferred into GV3101 chemically competitive cells (Weidi, Biotechnology, Shanghai, China). Before VIGS injection, pTRV1 and pTRV2 were mixed in equal volumes. Each treatment had 30 lettuce plants. Four treatment groups were established: blank control (no injection), empty vector (injection with equal pTRV2 and pTRV1), silencing (injection with equal pTRV2-*LsHSP70-19* and pTRV1), and overexpression (injection of pRI101-*LsHSP70-19*). Seven days after injection, high-temperature treatment was performed. After 14 days of treatment, three plants with consistent growth were selected from each treatment group, and new leaves were taken. DNA was extracted from each plant in triplicate to verify the positive transformation. Samples were collected from different parts of each plant in triplicate. RNA was extracted to assess vector expression, determine the expression level of *LsHSP70-19*, and measure the growth indexes of each group.

## 5. Conclusions

The 37 *LsHSP70* gene family members were identified from the whole genome of lettuce by the bioinformatics method, and their physicochemical properties, chromosomal localization, phylogenetic relationship, gene structure, and cis-acting element analysis of the promoter were analyzed. To understand the response of *LsHSP70* to high-temperature stress, we analyzed its expression patterns by transcriptome data. *LsHSP70-19*, with a more prominent expression pattern, was selected to preliminarily verify its positive regulatory role under high-temperature stress through virus-induced gene silencing and overexpression. The molecular function of *LsHSP70* under high-temperature stress requires further investigation.

## Figures and Tables

**Figure 1 ijms-26-00102-f001:**
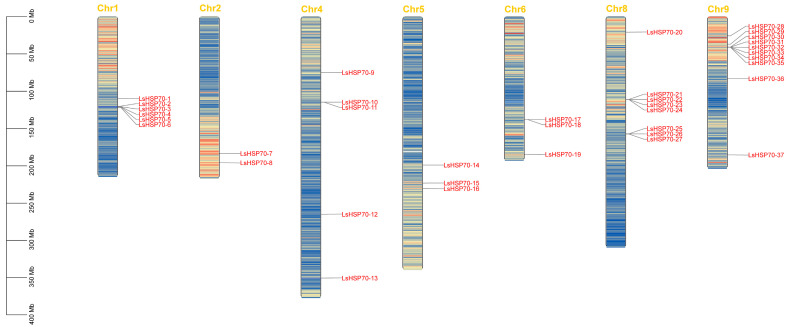
Chromosome localization of *LsHSP70*. The top yellow text represents the number of chromosomes, the red text represents the gene name, and the scale on the left represents the chromosome size.

**Figure 2 ijms-26-00102-f002:**
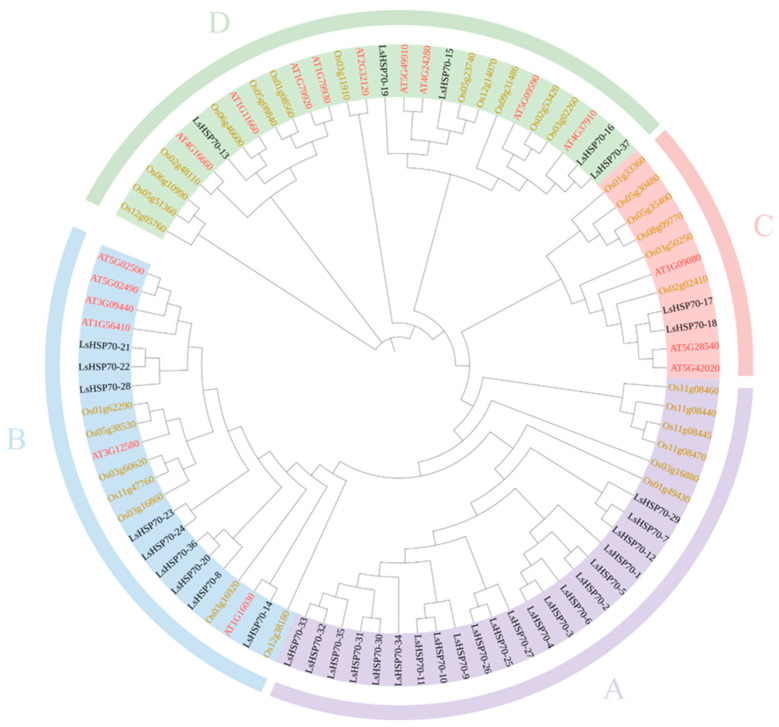
Phylogenetic tree of HSP70 gene family in lettuce, *Arabidopsis thaliana* and rice. Lettuce genes are shown in black, Arabidopsis genes are shown in red, and rice genes are shown in yellow.

**Figure 3 ijms-26-00102-f003:**
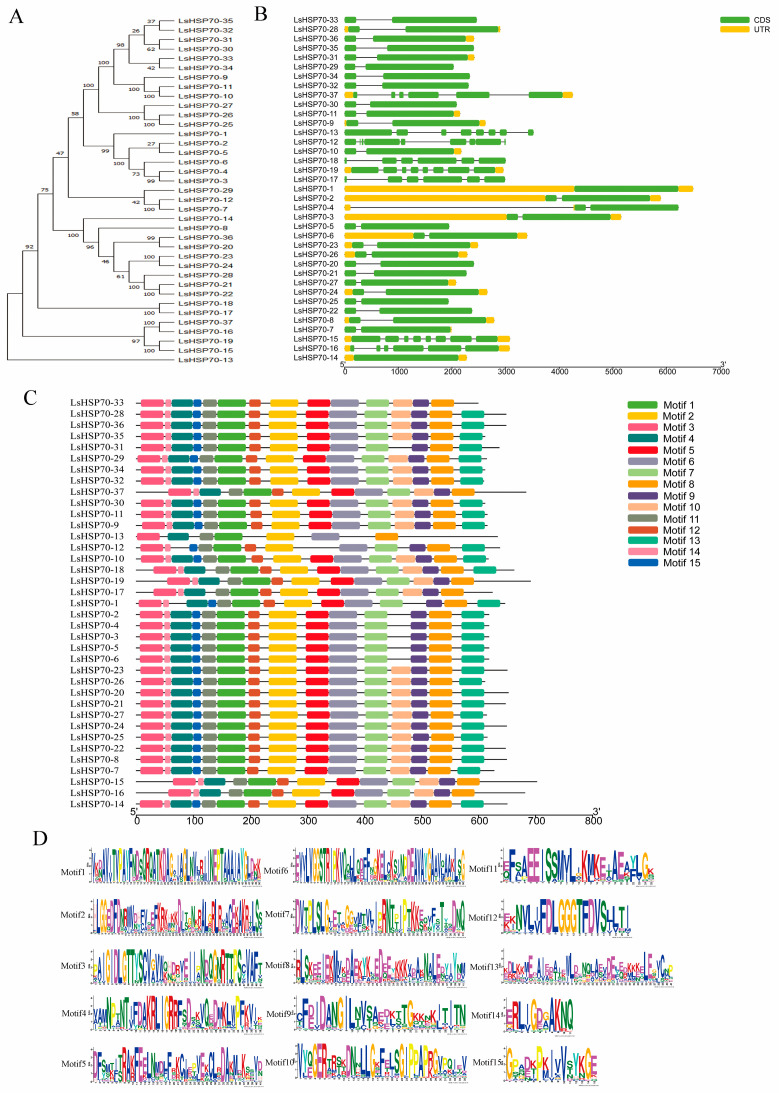
Gene structure of *LsHSP70* and motif analysis. (**A**) *LsHSP70* evolutionary tree; (**B**) gene structure; (**C**) motif distribution; (**D**) motif pattern. Figure 3D represents the sequence identity of the motif. Different colors correspond to different motifs, and different letter sizes represent different frequency matrices.

**Figure 4 ijms-26-00102-f004:**
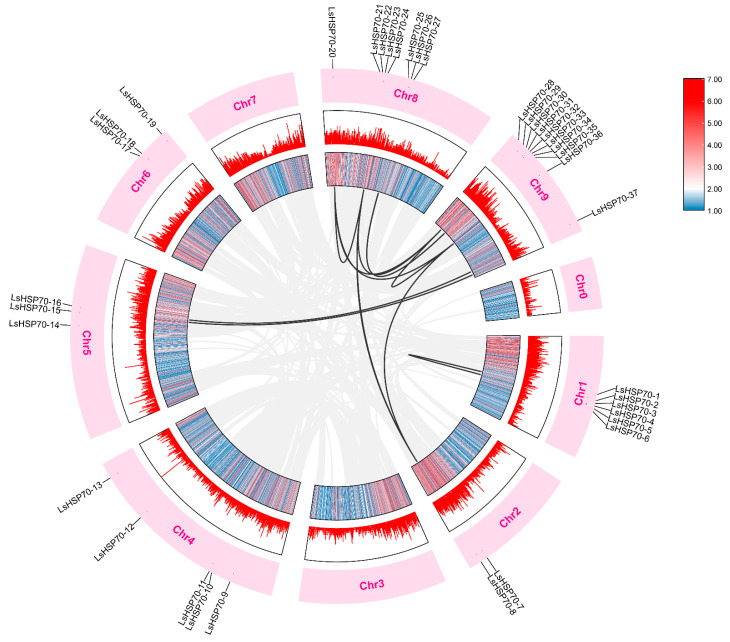
Collinearity analysis of *LsHSP70* gene family. The gray lines in the background represent homogenous pairs in the entire lettuce genome, and the black lines represent homogenous pairs in the *LsHSP70*. The information represented by each circle in the figure from the inside out was the genome density heat map, genome density linear map, chromosome name, and *LsHSP70* chromosome location labeling.

**Figure 5 ijms-26-00102-f005:**
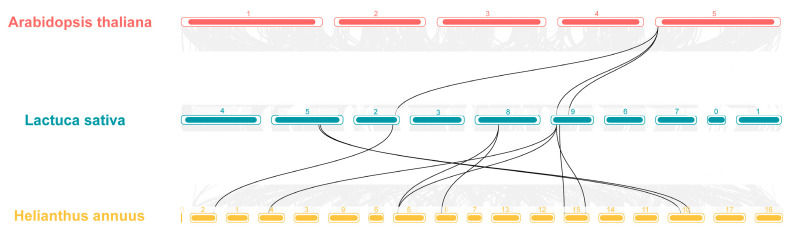
Collinearity analysis of lettuce with Arabidopsis and sunflower. The gray line in the background represents all collinear pairs of lettuce with Arabidopsis and sunflower at the genome level. The black line represents a common pair of HSP70 family members belonging to the three species.

**Figure 6 ijms-26-00102-f006:**
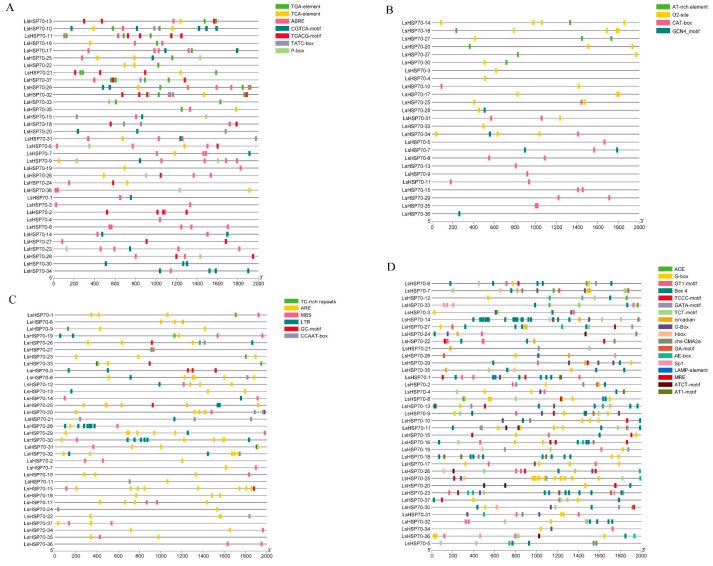
*LsHSP70* promoter analysis. Location map of cis-acting elements related to plant hormones (**A**), plant development (**B**), stress response (**C**), and light response (**D**). (**E**) Heat map of the number of instantaneous cis-acting components.

**Figure 7 ijms-26-00102-f007:**
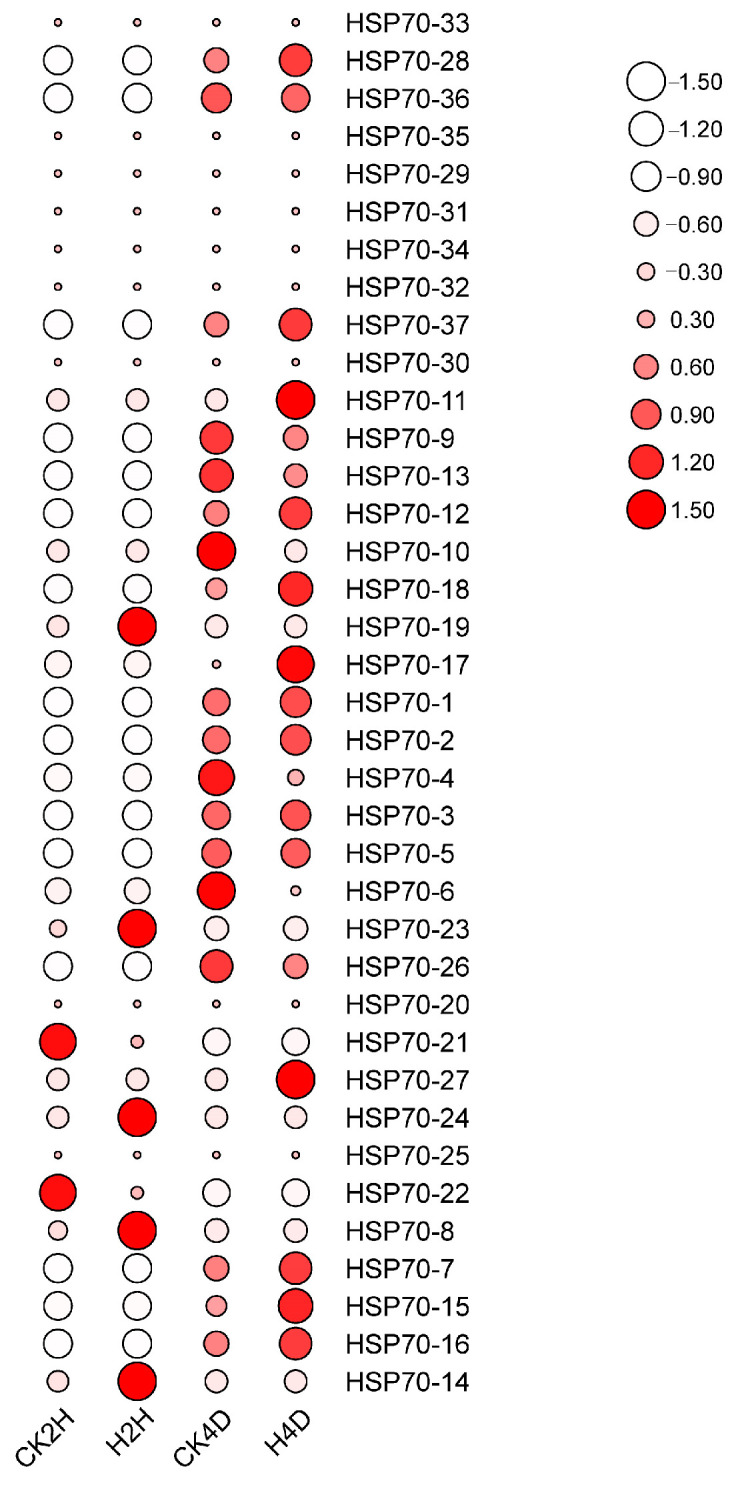
Expression of *LsHSP70* at high temperature. CK, room temperature treatment; H, high-temperature treatment; 2H, treatment two hours; 4D, treatment four days.

**Figure 8 ijms-26-00102-f008:**
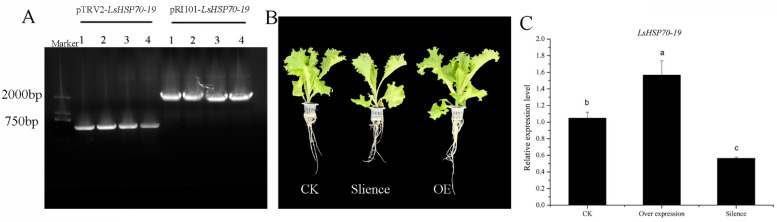
Role of *LsHSP70-19* at high temperature. (**A**) Cloning of *LsHSP70-19*; 1–4 in the figure represent 4 replicates. (**B**) Phenotypic map of lettuce under different treatments. (**C**) The expression of *LsHSP70-19* under different treatments. CK, control plants; Silence, silent plants; OE, overexpression plants. Different letters represent significant differences, determined by one-way ANOVA and Duncan’s test. *p* < 0.05.

**Table 1 ijms-26-00102-t001:** Physical and chemical properties of *LsHSP70*.

Gene Name	Locus ID	Chromosomal Location				Length (aa)	MW (kDa)	pI	Subcellular Localization
		Chr	Chr-Start	Chr-End	Direction				
*LsHSP70-1*	Lsat_1_v5_gn_1_87340	1	109,889,259	109,895,753	-	645	72	5.18	Endoplasmic reticulum
*LsHSP70-2*	Lsat_1_v5_gn_1_91700	1	120,103,665	120,109,556	+	617	69	8.11	Endoplasmic reticulum
*LsHSP70-4*	Lsat_1_v5_gn_1_93321	1	121,003,737	121,009,958	+	617	68	7.84	Endoplasmic reticulum
*LsHSP70-3*	Lsat_1_v5_gn_1_91141	1	120,849,862	120,855,133	+	617	69	6.51	Mitochondrion
*LsHSP70-5*	Lsat_1_v5_gn_1_91241	1	121,066,030	121,067,978	-	617	69	7.53	Endoplasmic reticulum
*LsHSP70-6*	Lsat_1_v5_gn_1_91381	1	121,849,518	121,855,607	+	617	69	8.48	Endoplasmic reticulum
*LsHSP70-8*	Lsat_1_v5_gn_2_116640	2	195,818,062	195,820,849	+	648	71	5.07	Nucleus
*LsHSP70-7*	Lsat_1_v5_gn_2_105720	2	183,329,797	183,331,788	+	626	70	5.30	Chloroplast
*LsHSP70-11*	Lsat_1_v5_gn_4_72600	4	114,729,310	114,731,461	+	614	69	8.01	Mitochondrion
*LsHSP70-9*	Lsat_1_v5_gn_4_50420	4	74,953,550	74,956,176	+	614	69	8.32	Chloroplast
*LsHSP70-13*	Lsat_1_v5_gn_4_173180	4	350,529,997	350,533,516	+	632	71	6.21	Mitochondrion
*LsHSP70-12*	Lsat_1_v5_gn_4_137301	4	264,836,010	264,839,010	-	636	70	8.28	Endoplasmic reticulum Mitochondrion
*LsHSP70-10*	Lsat_1_v5_gn_4_72580	4	114,717,746	114,719,920	+	616	69	8.20	Mitochondrion
*LsHSP70-15*	Lsat_1_v5_gn_5_104601	5	223,025,473	223,028,553	-	701	75	5.14	Chloroplast
*LsHSP70-16*	Lsat_1_v5_gn_5_110901	5	230,527,527	230,530,603	-	680	73	5.78	Mitochondrion
*LsHSP70-14*	Lsat_1_v5_gn_5_89440	5	198,889,223	198,891,499	+	649	71	5.18	Nucleus
*LsHSP70-18*	Lsat_1_v5_gn_6_83520	6	137,995,823	137,998,823	-	661	73	5.08	Endoplasmic reticulum Nucleus
*LsHSP70-19*	Lsat_1_v5_gn_6_112621	6	184,461,293	184,464,461	-	690	74	5.11	Chloroplast
*LsHSP70-17*	Lsat_1_v5_gn_6_83581	6	137,948,656	137,951,646	+	623	69	5.84	Endoplasmic reticulum Nucleus
*LsHSP70-23*	Lsat_1_v5_gn_8_78021	8	111,243,941	111,246,426	+	649	71	5.13	Nucleus
*LsHSP70-26*	Lsat_1_v5_gn_8_103521	8	156,999,462	157,001,748	-	610	68	6.47	Chloroplast
*LsHSP70-20*	Lsat_1_v5_gn_8_15681	8	20,742,298	20,744,707	+	651	72	5.58	Nucleus
*LsHSP70-21*	Lsat_1_v5_gn_8_78141	8	111,049,782	111,052,054	+	646	71	5.12	Chloroplast
*LsHSP70-27*	Lsat_1_v5_gn_8_103901	8	157,310,376	157,312,452	-	613	68	8.35	Mitochondrion
*LsHSP70-24*	Lsat_1_v5_gn_8_78000	8	111,278,374	111,281,033	-	648	71	5.15	Endoplasmic reticulum Mitochondrion
*LsHSP70-25*	Lsat_1_v5_gn_8_104441	8	156,970,950	156,973,039	+	614	68	8.47	Mitochondrion
*LsHSP70-22*	Lsat_1_v5_gn_8_78100	8	111,121,876	111,124,251	-	646	71	5.12	Chloroplast
*LsHSP70-33*	Lsat_1_v5_gn_9_36681	9	40,914,392	40,916,855	-	598	66	8.24	Chloroplast
*LsHSP70-28*	Lsat_1_v5_gn_9_22920	9	25,664,742	25,667,641	+	647	71	5.11	Chloroplast
*LsHSP70-36*	Lsat_1_v5_gn_9_65381	9	83,179,845	83,182,256	+	647	71	5.2	Chloroplast
*LsHSP70-35*	Lsat_1_v5_gn_9_36981	9	41,045,215	41,047,623	-	610	68	8.02	Chloroplast
*LsHSP70-31*	Lsat_1_v5_gn_9_36560	9	40,874,942	40,877,359	+	635	71	8.55	Endoplasmic reticulum Mitochondrion
*LsHSP70-29*	Lsat_1_v5_gn_9_34760	9	36,912,224	36,914,256	+	613	68	5.08	Mitochondrion Nucleus
*LsHSP70-34*	Lsat_1_v5_gn_9_36861	9	40,973,006	40,975,340	-	610	68	5.56	Chloroplast
*LsHSP70-32*	Lsat_1_v5_gn_9_36600	9	40,892,961	40,895,271	+	608	67	5.73	Chloroplast Nucleus
*LsHSP70-37*	Lsat_1_v5_gn_9_113681	9	185,253,863	185,258,114	-	682	73	5.89	Mitochondrion
*LsHSP70-30*	Lsat_1_v5_gn_9_36480	9	40,736,150	40,738,239	+	610	68	6.20	Chloroplast Nucleus

**Table 2 ijms-26-00102-t002:** Effects of *LsHSP70-19* transient expression or silencing on growth indices of lettuce.

Treatment	Fresh Weight (g)	Plant Height (cm)	Root Length (cm)	Shoot Weight (g)	Root Weight (g)
CK	14.49 ± 1.05 b	17.97 ± 2.40 a	21.6 ± 1.67 b	9.89 ± 0.84 b	2.34 ± 0.50 a
Overexpression	19.5 ± 1.22 a	21.67 ± 2.00 a	27.53 ± 2.21 a	17.31 ± 0.67 a	2.32 ± 0.92 a
Silence	10.44 ± 0.91 c	19.63 ± 0.78 a	27.67 ± 3.21 a	7.26 ± 0.12 c	1.66 ± 0.36 a

Note: Different letters represent significant differences determined by one-way ANOVA and Duncan’s test. *p* < 0.05.

## Data Availability

All of the data are contained within the article.

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
