# Peer review of "Genome-Wide Identification and Expression Analysis of HSP70 Gene Family Under High-Temperature Stress in Lettuce (Lactuca sativa L.)"

_ijms, 2024, doi:10.3390/ijms26010102_

Round 1
Reviewer 1 Report
Comments and Suggestions for Authors
The whole manuscript should be written more carefully:
“ The performed bioinformatics of the LsHSP70 gene family….”
“that most of the LsHSP70 genes were able to respond to heat stress, but only a few genes were able to respond rapidly to heat stress..”
“the overexpression plants showed better for heat tolerance”
Introduction:
Latin names of species should be in italic: Arabidopsis thaliana line 61 or Arabidopsis Thaliana line 116
“The expression pattern of LsHSP70 under heat stress at different 81 times were studied, the LsHSP70-19 gene, was responded rapidly to heat stress..”
Results:
“Referring to the structural sequence of Arabidopsis HSP70, 37 HSP70 genes in lettuce were mined. “
Could you please define “structural sequence”?
Or how “genes were enriched on chromosome 9”.
On the other hand these dense clusters of HSP genes seem t be quite interesting
Authors say that:
“It was suggested that these 6 Motifs may have a central role in the structural domain of LsHSP70” could you please expand on this thought?
There should be clear information in the results section 2.6, how expression data were obtained.
Better description/citations of vectors should be given.
“Total RNA extraction from lettuce leaves was performed by a kit..”?
The description of the biological part of experiment is unacceptably scarce, including methods section.
Comments on the Quality of English LanguageThe whole manuscript should be written more carefully:
“ The performed bioinformatics of the LsHSP70 gene family….”
“that most of the LsHSP70 genes were able to respond to heat stress, but only a few genes were able to respond rapidly to heat stress..”
“the overexpression plants showed better for heat tolerance”
Introduction:
Latin names of species should be in italic: Arabidopsis thaliana line 61 or Arabidopsis Thaliana line 116
“The expression pattern of LsHSP70 under heat stress at different 81 times were studied, the LsHSP70-19 gene, was responded rapidly to heat stress..”
Results:
“Referring to the structural sequence of Arabidopsis HSP70, 37 HSP70 genes in lettuce were mined. “
Could you please define “structural sequence”?
Or how “genes were enriched on chromosome 9”.
On the other hand these dense clusters of HSP genes seem t be quite interesting
Authors say that:
“It was suggested that these 6 Motifs may have a central role in the structural domain of LsHSP70” could you please expand on this thought?
There should be clear information in the results section 2.6, how expression data were obtained.
Better description/citations of vectors should be given.
“Total RNA extraction from lettuce leaves was performed by a kit..”?
The description of the biological part of experiment is unacceptably scarce, including methods section.
Reviewer 2 Report
Comments and Suggestions for Authors
The study focuses on the identification and analysis of the HSP70 gene family in lettuce, an agronomically important crop. Its emphasis on heat stress adds significant value, especially in light of the increasing impact of climate change on agriculture. However, several aspects should be addressed
a) Provide a more detailed comparison of how these findings in lettuce align with or differ from studies in other species (e.g., Arabidopsis, rice, cotton).
b) Figures 6 and 7 (promoter analysis and expression patterns) are informative but dense. Consider simplifying or breaking them into more readable subplots to improve accessibility.
c) Section 4.8 provides minimal details about the validation of gene constructs and qPCR confirmation. Adding more information on controls and success criteria would enhance reproducibility.
d) Some references are improperly formatted or lack details (e.g., references 15 and 29). These should be corrected to meet IJMS standards.
e) Although the manuscript mentions replicates and relative expression analysis, the statistical methods used (e.g., ANOVA, post-hoc tests) are not described. This should be addressed in the methods section or figure/table captions.
f) Refine the discussion with a stronger comparative análisis
g) Expand the methods section to ensure reproducibility.
Comments on the Quality of English LanguageThe manuscript contains multiple grammatical errors and stylistic issues, which affect readability. For example: "It was suggested that overexpression of LsHSP70-19 would improve the heat tolerance of lettuce" (page 11). This should be rephrased for clarity and to avoid redundancy.
Round 2
Reviewer 1 Report
Comments and Suggestions for Authors
The information about sequences of vectors used for plant transformation with A. tumefaciens: pTRV1, pTRV2 and pRI101 is still missing.
Author Response
Comments 1: [The information about sequences of vectors used for plant transformation with A. tumefaciens: pTRV1, pTRV2 and pRI101 is still missing.]
Response 1: Thank you for pointing this out. We have added to this.
[We have uploaded pTRV1, pTRV2 and pRI101 vector sequences to "Non-published Material". In addition, we also added the primer sequence "Table S1" for the experiment and uploaded it to the "Supplementary File(s)".
We also made a change to the author ranking. "Qian Wang 1,‡, Wenjing Sun 1,‡, Yipei Duan 1, Yikun Xu 2, Huiyu Wang 1, Jinghong Hao 1, Chaojie Liu 1,* and Yingyan Han 1,* "is changed to" Qian Wang 1,‡, Wenjing Sun 1,‡, Yipei Duan 1, Yikun Xu 2, Huiyu Wang 1, Jinghong Hao 1, Yingyan Han 1 and Chaojie Liu 1,* ", the changes made are shown in yellow in the manuscript.]
PS:
Prof. Dr. Yingyan Han agrees to change from corresponding author to co-author. All the authors agree.